# Neurocognitive and mental health outcomes and association with quality of life among adults living with HIV: a cross-sectional focus on a low-literacy population from coastal Kenya

Moses Kachama Nyongesa,[1] Patrick N Mwangala,[1] Paul Mwangi,[1] Martha Kombe,[1] Charles R J C Newton,[1,2,3] Amina A Abubakar[1,2,3]

[1]Neuroassessment Group, KEMRI-Wellcome Trust Research Programme, Center for Geographic Medicine Research (Coast), Kilifi, Kenya
[2]Department of Psychiatry, University of Oxford, Oxford, UK
[3]Department of Public Health, Pwani University, Kilifi, Kenya

**Correspondence to**
Moses Kachama Nyongesa;
Mkachama@kemri-wellcome.org

## ABSTRACT

**Objectives** Our aim was to compare the neurocognitive performance and mental health outcome of adults living with HIV on antiretroviral therapy with that of community controls, all of low literacy. Furthermore, we also wanted to explore the relationship of these outcomes with quality of life among adults living with HIV.

**Study design** This was a descriptive cross-sectional study.

**Setting** The study was conducted in Kilifi County, a region located at the Kenyan coast.

**Participants** The participants consisted of a consecutive sample of 84 adults living with HIV and 83 randomly selected community controls all with ≤8 years of schooling. All participants were assessed for non-verbal intelligence, verbal working memory and executive functioning. The Major Depression Inventory and a quality of life measure (RAND SF-36) were also administered.

**Results** Using analysis of covariance, we found no statistically significant group differences between adults living with HIV and community controls in all the neurocognitive tests except for a marginal difference in the non-verbal intelligence test ($F_{(1, 158)}=3.83$, p=0.05). However, depressive scores of adults living with HIV were significantly higher than those of controls ($F_{(1, 158)}=11.56$, p<0.01). Also, quality of life scores of adults living with HIV were significantly lower than those of controls ($F_{(1, 158)}=4.62$, p=0.03). For the HIV-infected group, results from multivariable linear regression analysis showed that increasing depressive scores were significantly associated with poorer quality of life ($\beta=-1.17$, 95% CI −1.55 to −0.80; p<0.01).

**Conclusion** Our findings suggest that adults of low-literacy levels living with HIV and on antiretroviral medication at the Kenyan coast do not have significant cognitive deficits compared with their uninfected counterparts. However, their mental health, compared with that of HIV-uninfected adults, remains poorer and their quality of life may deteriorate when HIV and depressive symptoms co-occur.

## Strengths and limitations of this study

▶ This is one of the few studies simultaneously exploring cognitive, mental health and quality of life outcomes among HIV-infected sample from sub-Saharan Africa.
▶ We also focus specifically on a low-literacy sample.
▶ The key limitation is the cross-sectional nature of our study design, therefore, we cannot ascertain any causal relationship.

## INTRODUCTION

The introduction of highly active antiretroviral therapy (HAART)—more than three decades now—has resulted to significant declines in HIV/AIDS-related morbidity and mortality.[1] An increase in life expectancy among individuals living with HIV and on HAART has also been observed in both low-resourced[2 3] and high-resourced[4] settings. For patients with HIV who are optimally treated with antiretroviral medication, their life expectancy may approach that of HIV-uninfected population.[5] Despite this achievement in longevity, neurocognitive disturbance and psychological problems remain as important health issues of concern among people living with HIV/AIDS (PLWH).[6 7]

Neurocognitive impairment (NCI) is a common complication following HIV infection,[8 9] even with the advent of HAART.[10] The NCI associated with HIV, also known as HIV-associated neurocognitive disorder (HAND), arises when HIV-infected monocytes and macrophages infiltrate the brain by crossing the blood–brain barrier, to infect and activate microglia causing inflammation of the central nervous system and progressive cognitive dysfunction (from involvement of the cortical and subcortical regions).[9 11] In

sub-Saharan Africa (SSA), where there is a substantial burden of HIV infection compared with other regions of the world, prevalence estimates for HAND as high as 42% (pre-HAART) and 30% (post-HAART) have been reported.[12] Some of the cognitive deficits described among PLWH, through neuropsychological testing, include impairment in attention/concentration, processing speed and executive functioning, as well as problems with motor functioning, learning and memory.[13–15] For PLWH on HAART and with well-controlled viraemia, mild NCI is much more common than severe cognitive impairment, that is, HIV-associated dementia.[10 16] There are significant clinical and functional consequences of having even mild NCI including: increased risk of mortality[17]; poor medication adherence[18]; decreased quality of life[19] and disruptions to everyday functioning.[20]

Mental disorders are also common among PLWH, depression being the most prevalent.[21 22] A review of the mental health of adults living with HIV in Africa[21] reported estimates for depressive symptoms of over 30% and as high as 64% from different reviewed studies. Furthermore, major depressive disorder has been found to occur almost twice as frequent among PLWH compared with the HIV-uninfected subjects.[23]

There are many reasons as to why PLWH may experience mental disorders frequently, particularly depressive symptoms. First, is the difficulty of living with a chronic, life-threatening illness. This adjustment difficulty—presenting as psychological distress—may occur after receiving HIV positive diagnosis; when HIV symptoms begin to develop; at the start of lifelong antiretroviral medication and also in terminal care.[24] Second, the direct effects of HIV virus on the central nervous system[25] may lead to behavioural changes resulting into symptoms of depression. Third, such psychological disturbance may also be because of HIV-related stigma.[26] PLWH with comorbid mental illnesses may present with poor quality of life[22] and non-adherence to antiretroviral medication.[27] Non-adherence may lead to faster disease progression or increased HIV-related/AIDS-related mortality.

Cognitive and mental health outcomes of PLWH and their relationship with quality of life have gained increasing attention recently, with most work being conducted and reported from highly resourced countries, but limited (although increasing) reports from SSA, where the burden of HIV/AIDS is substantial. To add to the body of research evidence from SSA, we evaluated the neurocognitive test performance and mental health of adults living with HIV on antiretroviral medication compared with community controls, specifically focusing on a low-literacy population at the Kenyan coast. Less education among PLWH has been associated with an increased risk for symptomatic HAND.[28 29] We also set out to explore the relationship between each of these outcomes with quality of life among adults living with HIV.

## METHODS

### Study site
This study was undertaken at the Centre for Geographic Medicine Research-Coast (CGMR-C), Kilifi County, Kenya between November 2016 and March 2017. The Centre has a neuroassessment unit equipped for assessing a wide range of neurocognitive and neuropsychiatric outcomes in children, adolescents and adults living with chronic illnesses such as HIV and epilepsy. Interviews and tests are carried out by trained and experienced research assistants. In Kilifi County, the estimated overall adult HIV prevalence is 4.5% with prevalence being higher among females (6.4%) than males (2.7%).[30]

### Study design
This was a cross-sectional study aimed at investigating neurocognitive functioning, mental health and their association with quality of life in an adult sample. This study was embedded in a larger study whose aim was to validate a tablet-based android application (App)—the NeuroScreen, Swahili version—screening for NCI among chronically ill adults in Kilifi. In the NeuroScreen study, only participants with ≤8 years of schooling (up to primary level of education in the Kenyan education system) were included with the assumption that if the App works with the less literate individuals in Kilifi, then it should work well with those who are better educated.

### Sample size estimation
In this current study, the sample size (n=167) included 84 participants living with HIV and 83 community controls. This sample size was calculated to detect differences in cognitive outcomes between adults living with HIV and controls with 90% power using previously reported effect sizes.[31 32]

### Recruitment and criteria for inclusion
#### Participants living with HIV
Adults living with HIV were recruited from the Comprehensive Care and Research Clinic, an HIV specialised clinic at the Kilifi County Hospital. Selection and booking for assessment date was done through consecutive sampling of clients on their arrival at the clinic. Participants were eligible for inclusion in the study if they: (1) were 20–50 years old; (2) had a confirmed diagnosis of HIV; (3) had no more than primary level of education (an inclusion criterion for the larger study validating the NeuroScreen); (4) could understand and communicate in Kiswahili (the official national language of Kenya) with ease; (5) provided informed consent and (6) had no obvious symptoms of acute illness on the day of assessment.

#### Community controls
Community controls were identified through the Kilifi Health and Demographic Surveillance System (KHDSS) by random selection and later approached at home and requested to take part in the study. Details about the KHDSS can be found elsewhere.[33] Those who consented

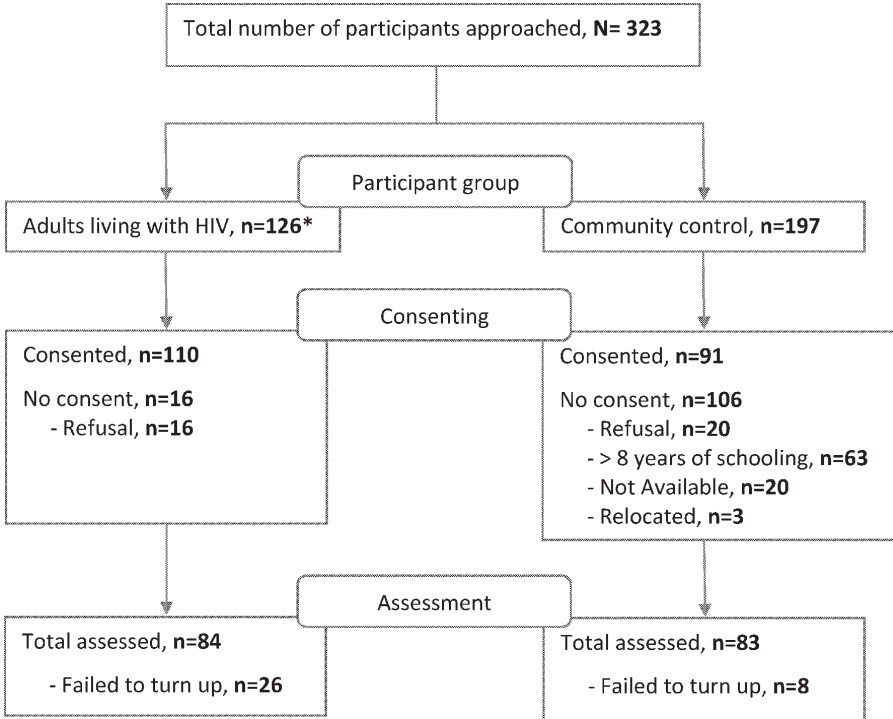

Total number of participants approached, **N= 323**

Participant group

Adults living with HIV, **n=126\*** | Community control, **n=197**

Consenting

Consented, **n=110**

No consent, **n=16**
- Refusal, **n=16**

Consented, **n=91**

No consent, **n=106**
- Refusal, **n=20**
- > 8 years of schooling, **n=63**
- Not Available, **n=20**
- Relocated, **n=3**

Assessment

Total assessed, **n=84**

- Failed to turn up, **n=26**

Total assessed, **n=83**

- Failed to turn up, **n=8**

**Notes**: * Identified as having ≤8 years of schooling

**Figure 1** Flow chart of participant recruitment.

to taking part in the study were referred to CGMR-C for assessment if they met the eligibility criteria as above, except that of having an HIV diagnosis. These participants had to self-report a seronegative test result in the last year. We did not test for HIV among these controls, and we acknowledge that in randomly selecting them, about 3 or 4 out of the 84 may have been HIV infected considering the adult HIV prevalence in Kilifi County, Kenya.[30] However, through a detailed clinical examination (see Study procedures section), we only included participants having no symptoms of an acute illness. Using this criterion, we expected very few cases of asymptomatic HIV infection among our controls that is unlikely to invalidate our findings.

Figure 1 is a flow chart showing the recruitment process for both adults living with HIV and community controls in line with the Strengthening the Reporting of Observational Studies in Epidemiology (STROBE) guidelines.[34]

### Study procedures

Every consented participant underwent a detailed medical history and symptomatology clinical examination. For controls, other than ruling out symptoms of an obvious acute illness for participation, symptomatological evaluation was also done to rule out the possibilities of HIV infection. A set of questionnaires were then administered to capture information about sociodemographic and economic background, mental health and quality of life. Cognitive assessments were paper and pencil and computer tablet based. Data collection was conducted by a trained team of research assistants.

### Measures

The measures listed below were all administered to both adults living with HIV and community controls. For both groups, all measures were administered in the same order, by the same assessor (who was blinded) and under the same administration environment.

### Social demographic and asset index forms

These were used to collect information on participant's age, sex, marital status, level of education, and socioeconomic status. The asset index has been reliably used to assess socioeconomic status over time in studies conducted in this setting.[35 36] The asset index items ask for ownership of disposable assets by participants (or their family) such as radio, television, bicycle or motorbike.

### Major Depression Inventory

This is a self-report measure of depressive symptoms that examines moods experienced by an individual during the past 2 weeks. It consists of 12 items, although only 10 items are scored since for items 8 (a and b) and 10 (a and b) only an alternative with the highest score is considered. Items are scored on a 6-point Likert scale from 0 (at no time) to 5 (all the time) with the total score ranging from 0 to 50.[37] The Major Depression Inventory (MDI) can be used both as a diagnostic instrument for major depression according to the Diagnostic and Statistical Manual of Mental Disorders, version four and as a depression rating scale. As a rating scale, MDI total score of 20–24 is indicative of mild depressive symptoms, a score of 25–29 is indicative of moderate depressive symptoms whereas

a score ≥30 indicates severe depressive symptoms.[38] This measure, adapted to Swahili, yielded reliable psychometric properties among young adults in Kilifi.[39]

### Measures of cognitive performance

In this study, three tests were administered to assess cognitive functioning among study participants, that is, the Raven's Standard Progressive Matrices (RSPM),[40] contingency naming test (CNT)[41] and the backward digit span test.[42] RSPM is a non-verbal standardised test of IQ comprising five series with 12 items without limit in execution time. The CNT, initially developed to assess aspects of executive functioning such as processing speed and response inhibition in children, has shown good psychometric reliability when extended to the adult population.[43] In the CNT test, a participant is required to name the colour or shape of a series of stimuli, as fast and accurately as possible, according to different rules. The backward digit span test assesses verbal working memory.[42] A list of random numbers is read aloud to the participant who is then required to recall the items in a reverse order.

### The RAND 36-Item Self-report form V.1.0 (RAND SF-36)

This is the most widely used generic measure of health-related quality of life. It assesses eight health concepts: physical functioning, role limitations due to physical health problems, role limitations due to emotional problems, social functioning, emotional well-being, energy/fatigue, pain and general health perceptions.[44] Scoring the tool involves various steps. First, precoded numeric values are converted to percentages using a predetermined scoring key. Second, items in the same scale are summed and then averaged to give the 8 different subscale scores. Lastly, to generate a total score, all item scores are summed and then averaged. Higher scores (as percentage) define a more favourable state of health. RAND SF-36 has been found to be reliable in SSA.[45] For this study, the English questionnaire was forward translated into Kiswahili by two independent translators. Back translation into English was then undertaken by two other independent back translators (unaware of the original version). The final version was obtained following a harmonisation process by a team of experts and incorporation of changes resulting from a cognitive interview process and pretesting procedures. In the current study, Cronbach's alpha for this measure was excellent at 0.93.

### Patient and public involvement

Patients were not involved in the design and conduct of the study.

### Statistical analyses

All analyses were carried out on STATA (V.14.0) statistical software package (StataCorp).[46] Frequencies, percentages, mean or median were used to describe sample characteristics. $\chi^2$ tests (with Fisher's exact test for continuity correction where appropriate) and independent Student's t-test (or Wilcoxon rank-sum test where appropriate) were used to compare group differences between

categorical and continuous variables, respectively. Analysis of covariance (ANCOVA) was used to control for covariates of interest when looking at group differences in neurocognitive test performance, depressive and quality of life scores between adults living with HIV and community controls. We checked for ANCOVA's homogeneity of regression slope assumption using a method suggested by Johnson.[47] In an analysis involving only the HIV-infected group, we used correlational and multivariable linear regression analyses to explore potential associations between neurocognitive test performance, depressive scores and quality of life scores. All tests in our analyses were considered statistically significant at p<0.05.

## RESULTS

### Sample characteristics

Table 1 below describes sample characteristics by HIV status (ie, adults living with HIV vs controls). The mean age of the 167 study participants who took part in the study was 36.95 (SD=8.65). Majority of the study participants were: females (71.3%) and married (63.9%). Less than a quarter of study participants (16.4%) did not have any formal education. As shown in table 1, adults living with HIV were significantly different from controls in all sociodemographic characteristics except for socioeconomic status (p=0.92). Nutritional status of the study participants (as indicated by body mass index and mid-upper arm circumference) and the clinical information of adults living with HIV are also presented in table 1. We also present observations with missing values.

### Neurocognitive performance, mental health and quality of life of adults living with HIV compared with community controls

#### Neurocognitive performance

In the crude analysis, adults living with HIV compared with controls had significantly lower mean non-verbal IQ scores as measured by the RSPM (p<0.01; table 2) and verbal working memory scores, that is, lower mean total correct scores (p=0.01; table 2) and mean score of highest set of digits reached (p<0.01; table 2) of the backward digit span test. However, no significant difference was observed between adults living with HIV and controls in executive functioning as assessed by CNT total error scores (p=0.32; table 2). Using ANCOVA adjusting for sex, age, level of education and socioeconomic index as covariates, there were no significant differences between adults living with HIV and community controls for all neurocognitive test scores except non-verbal IQ scores where a marginal difference was observed, F (1, 158)=3.83, p=0.05, with a small effect size (0.29; table 2).

#### Mental health

Twenty (12.0%) of study participants met the criteria for mild to severe depressive symptoms (15 being PLWH and 5 being community controls) as measured by the MDI. Adults living with HIV had significantly higher mean depressive scores than controls (p<0.01; table 2) in the crude analysis.

**Table 1** Sample characteristics of adult participants living with HIV and community controls (n=167) along with clinical characteristics of participants living with HIV

| Sample characteristics | Total sample | HIV infected n=84 | Controls n=83 | P values |
|---|---|---|---|---|
| | n (%) or mean (SD) or median (IQR) | | | |
| Age (in years; mean (SD)) | 37.0 (8.7) | 40.1 (6.4) | 33.8 (9.5) | **<0.01*** |
| Sex | | | | |
| Female | 119 (71.3) | 66 (78.6) | 53 (63.9) | **0.04** |
| Male | 48 (28.7) | 18 (21.4) | 30 (36.1) | |
| Marital status OM=1 | | | | |
| Never married | 27 (16.3) | 8 (9.5) | 19 (23.2) | **<0.01†** |
| Married | 106 (63.9) | 47 (56.0) | 59 (72.0) | |
| Separated/divorced/widowed | 33 (19.9) | 29 (34.5) | 4 (4.9) | |
| Education OM=2 | | | | |
| None | 27 (16.4) | 20 (24.1) | 7 (8.5) | **<0.01** |
| Primary incomplete | 73 (44.2) | 47 (56.6) | 26 (31.7) | |
| Primary complete | 65 (39.4) | 16 (19.3) | 49 (59.8) | |
| SES Asset Index (median (IQR)) | 2.0 (1.0–3.0) | 1.0 (1.0–3.0) | 2.0 (1.0–2.0) | 0.92‡ |
| BMI (median (IQR)) | 22.4 (20.2–26.2) | 22.4 (19.0–26.7) | 22.3 (20.5–25.9) | 0.67‡ |
| MUAC (median (IQR)) | 26.9 (25.0–29.3) | 26.7 (24.2–29.0) | 27.1 (25.2–29.4) | 0.33‡ |
| Severity of depressive symptoms | | | | |
| No depressive symptom | 147 (88.0) | 69 (82.1) | 78 (94.0) | 0.06* |
| Mild depressive symptoms | 9 (5.4) | 8 (9.5) | 1 (1.2) | |
| Moderate depressive symptoms | 6 (3.6) | 4 (4.8) | 2 (2.4) | |
| Severe depressive symptoms | 5 (3.0) | 3 (3.6) | 2 (2.4) | |
| HIV-related clinical characteristics | | | | |
| HIV disease staging OM=6 | | | | |
| Stage 1 | NA | 38 (48.7) | NA | NA |
| Stage 2 | NA | 27 (34.6) | NA | NA |
| Stage 3 | NA | 13 (16.7) | NA | NA |
| No of ARV medication OM=7 | | | | |
| Two | NA | 56 (72.7) | NA | NA |
| Three | NA | 21 (27.3) | NA | NA |
| Duration on ARV | | | | |
| <1 year | NA | 4 (5.1) | NA | NA |
| 1–5 years | NA | 29 (37.2) | NA | NA |
| 6–10 years | NA | 36 (46.2) | NA | NA |
| >10 years | NA | 9 (11.5) | NA | NA |

Bold, statistically significant results (p-values).
Notes: Percentages are exclusive of missing values. For observations with missing values, the number of missing values is indicated in the table as OM (observations with missing values). P values are based on $\chi^2$ statistic except for:
*Based on independent t-test.
†$\chi^2$ based on Fisher's exact test.
‡Based on Mann-Whitney U test.
 ARV, antiretroviral; BMI, body mass index; CNT, contingency naming test; MUAC, mid-upper arm circumference; NA, not applicable; QoL, quality of life; SES, socioeconomic status.

ANCOVA controlling for sex, age, education and socio-economic status (table 2) still found significant differences between adults living with HIV and community controls, F $(1, 158)=11.56$, $p<0.01$.

**Quality of life**
In the crude analysis, adults living with HIV compared with controls had significantly lower mean quality of life scores as measured by the RAND SF-36 (p<0.01; table 2).

**Table 2** Unadjusted and adjusted analyses for neurocognitive test performance, mental health and quality of life outcomes comparing adults living with HIV and community controls

| Outcome | Unadjusted analysis | | | Adjusted analysis* | | | | Eta-squared (%) | ES |
|---|---|---|---|---|---|---|---|---|---|
| | HIV score | Control score | P values | HIV score | Control score | F - statistic | P values | | |
| Neurocognitive test performance | | | | | | | | | |
| CNT total errors | 2.82 (0.92) | 2.68 (0.85) | 0.32 | 2.89 (0.12) | 2.79 (0.12) | 0.38 | 0.54 | 6.5 | 0.09 |
| Ravens total score | 16.82 (5.39) | 19.81 (7.28) | <0.01 | 16.72 (0.82) | 18.92 (0.83) | 3.83 | **0.05** | 14.6 | 0.29 |
| Digit span—total correct score | 5.06 (2.81) | 6.41 (3.32) | **0.01** | 4.85 (0.39) | 5.42 (0.41) | 1.05 | 0.31 | 18.7 | 0.16 |
| Digit span—highest set of digits reached† | 2.19 (1.26) | 2.90 (1.44) | <0.01 | 2.06 (0.17) | 2.47 (0.18) | 3.05 | 0.08 | 21.9 | 0.27 |
| Mental health | | | | | | | | | |
| Depressive scores | 11.80 (8.38) | 7.96 (7.32) | <0.01 | 12.23 (1.03) | 7.46 (1.04) | 11.56 | **<0.01** | 11.71 | 0.50 |
| Quality of life (QoL) | | | | | | | | | |
| QoL total score | 70.38 (17.18) | 78.07 (14.72) | <0.01 | 72.56 (2.12) | 78.78 (2.14) | 4.62 | **0.03** | 9.64 | 0.32 |

Bold, statistically significant results (p-values).

Notes: All scores presented as mean (SD), unadjusted analysis; and mean (SE), adjusted analysis; ES—Cohen's d effect size.

*ANCOVA adjusted for sex, age, level of education and socioeconomic index.

†There were 8 sets with a series of random digits; sets 1, 2 and 3 had a length of 3 digits, sets 4 through 8 had digit lengths of 4 through 8, respectively.

ANCOVA, analysis of covariance; CNT, contingency naming test.

Controlling for sex, age, education and socioeconomic status in ANCOVA (table 2), there was still a statistically significant difference between adults living with HIV and community controls, F (1, 158)=4.62, p=0.03.

### Correlation of neurocognitive, depressive and quality of life scores in the HIV-infected group

There was no statistically significant correlation between any of the neurocognitive scores and quality of life scores, that is, total correct score (r=−0.05, p=0.63), score of the highest set of digits reached (r=−0.01, p=0.96), CNT's total error scores (r=0.08, p=0.45) or Raven's total score (r=−0.05, p=0.68). However, correlation analysis revealed a significant correlation between depressive and quality of life scores (r=−0.67; p<0.01). The reported results in the subsequent sections will therefore focus on findings from analyses that were carried out based on the observed significant correlation between depressive and quality of life scores.

### Association between depressive symptoms and quality of life in the HIV-infected group

Exploring further the observed significant correlation between depressive and quality of life scores among adults living with HIV, a one-unit increase in depressive score was associated with 1.2 reduction in quality of life scores in the univariate linear regression analyses (p<0.01; table 3). Adjusting for age, sex, number of and duration on antiretroviral medication and HIV disease staging, there was still strong evidence of an association between elevated depressive scores and poorer quality of life (β=−1.17, 95% CI −1.55 to −0.80; p<0.01; table 3).

### DISCUSSION

In this study, we were interested in evaluating neurocognitive and mental health outcomes of adults living with HIV compared with community controls, specifically focusing on a low-literacy sample from the Kenyan coast. We also wanted to explore the relationship between each of these outcomes and quality of life among the HIV-infected adult group.

### Neurocognitive test performance

In summary, we did not observe main effect of HIV infection on most of the neurocognitive tests except that of non-verbal intelligence when comparing adults living with HIV and community controls all with lower educational background. For non-verbal intelligence, the adjusted group difference was marginally significant with a small effect size (0.3). We think that this marginal difference may be due to impaired premorbid intellectual functioning. Our findings generally contrast findings from other studies conducted in SSA. A Nigerian study by Sunmonu et al,[32] assessing intellectual performance of antiretroviral-naïve HIV-infected participants compared with a control group, found strong significant differences (p<0.01) with medium to large effect sizes (0.7–1.8).

**Table 3** Univariate and multivariable linear regression analyses for the association between elevated depressive scores (exposure) and poorer quality of life (outcome) among adults living with HIV in coastal Kenya, n=72

| Exposure variable | Univariate analysis | | | Multivariable analysis | | |
|---|---|---|---|---|---|---|
| | Unadjusted β coefficient (95% CI) | P values | R² (%) | Adjusted β coefficient* (95% CI) | P values | R² (%) |
| Depressive scores | −1.17 (−1.53 to 0.80) | **<0.01** | 36.70 | −1.17 (−1.55 to 0.80) | **<0.01** | 45.19 |
| Adjusted for: | | | | | | |
| Age | | | | −0.20 (−0.77 to 0.37) | 0.48 | |
| Sex (male vs female) | | | | 7.02 (−1.18 to 15.22) | 0.09 | |
| No of ARV medication (3 vs 2) | | | | −4.49 (−12.32 to 3.33) | 0.26 | |
| HIV staging | | | | | | |
| Stages 2 vs 1 | | | | −2.63 (−10.04 to 4.77) | 0.48 | |
| Stages 3 vs 1 | | | | −8.60 (−18.12 to 0.91) | 0.08 | |
| Duration on ARV | | | | | | |
| 1–5 years vs <1 year | | | | 7.83 (−6.63 to 22.29) | 0.28 | |
| 6–10 years vs <1 year | | | | 9.29 (−5.16 to 23.73) | 0.20 | |
| >10 years vs <1 year | | | | 11.28 (−4.87 to 27.43) | 0.17 | |

Bold, statistically significant results (p-values)
Notes: *Linear regression adjusted coefficient for quality of life.
ARV, antiretroviral.

On the other hand, a study conducted in Uganda by Robertson et al[31] reported significant differences between HIV-infected subjects and controls for the backward digit span test (p<0.001) and tests of executive functioning (<0.01).

There are two potential explanations for these differences in findings. First, we recruited study participants (both the HIV infected and controls) with ≤8 years of schooling unlike in the Nigerian and Ugandan studies[31 32] where participants had up to 13 years of schooling. Using this criterion, we selected a group that was potentially at an elevated risk of poor performance on neurocognitive tests.[48] It is not surprising to see that the mean scores of our study participants compared with participants in the Ugandan study,[31] for instance, were generally lower on the backward digit span test, that is, 2.2 vs 3.0 (for the HIV-infected groups) and 2.9 vs 3.5 (for the control groups). Even though our sample was of lower educational background, HIV-infected adults were significantly less literate than community controls. Accounting for the between-group educational differences among other factors, the initially observed unadjusted strong differences between adults living with HIV and community controls in non-verbal IQ and working memory tests disappeared.

Second, it may be plausible that our adult participants living with HIV were stable on ART. Only 4 participants out of the 78 with information on ART duration had used it for less than a year. ART is thought to delay the onset of milder NCI[49] (common even in the era of HAART)[10] or to improve the performance of patients with an existing HIV-associated neurocognitive dysfunction.[50] Either way, being stable on ART may have resulted in comparable neurocognitive test performance between adults living

with HIV and community controls in our sample and ART-naivety in the study by Sunmonu et al[32] may have explained why the authors observed strong significant group differences.

### Mental health
In this study, living with HIV was significantly associated with increased depressive scores even after adjusting for covariates such as age, sex, education and socioeconomic status between the HIV-infected and control groups. Our finding corroborates findings reported from studies conducted in South Africa[51] and Nigeria.[52] Less education—a proxy for low socioeconomic status—has been found to be a risk factor for common mental disorders in the general population.[53 54] In this study, we found meaningful adjusted differences in depressive scores comparing a sample of HIV-infected adults and community controls all with an already limited educational background. Our finding therefore suggests that a tendency towards scores indicative of depressive symptomatology may be because of HIV infection than a socioeconomic influence.

### Neurocognitive, mental health and quality of life among adults living with HIV
Among adults living with HIV, we found no correlation between any of the neurocognitive scores and quality of life scores. However, depressive scores strongly correlated with quality of life scores and further exploration using multivariable linear regression analyses showed that high depressive scores were associated with poorer quality of life. This kind of association has previously been reported from studies conducted in other low-to-middle income settings.[22 55] This finding suggests that interventions aimed at improving mental health are likely to contribute

to enhanced quality of life among adults living with HIV in our setting.

## Study limitations

This study has some limitations which should be considered when interpreting the findings. First, as a cross-sectional study design, we cannot ascertain any causal relationship for the observed significant associations. Second, recruiting community controls based on their self-report of HIV seronegative status may have introduced some bias. Third, because the larger study was primarily focused on psychometric evaluation of the neuroscreen, clinical information of biomarkers, that is, viral load and cluster of differentiation 4 (CD4) cell count for adults living with HIV was not available for the current study and consequently not accounted for in our analyses. Relatedly, HIV disease staging, for which we have considered in our analysis, was collected historically (last known disease stage) and may not be reflective of a participant's disease state at the time of this study. Lastly, we recruited and administered neurocognitive tests to a low-literacy sample. The potential lack of familiarity with neurocognitive test requirement/demands by our study participants, because of this biased selection, limits the generalisability of our finding. Our findings should be interpreted in the context of a low-literacy population.

## CONCLUSION

Our findings suggest that adults of low-literacy levels living with HIV and on antiretroviral medication at the Kenyan coast did not have cognitive deficits when compared with their uninfected counterparts. However, the mental health of HIV-infected adults remains poorer and their quality of life may deteriorate when HIV and depressive symptoms co-occur. Interventions to enhance early identification and management of depression may promote better quality of life of, especially the socioeconomically disadvantaged patients living with HIV in our setting and other similar settings.

**Acknowledgements** We wish to thank all our study participants for taking time, voluntarily, to participate in this study. We would also like to thank Katana Ngombo, Richard Karisa, Beatrice Kabunda, Judith Tumaini Dzombo and Khamisi Katana for their role in data collection. The authors would also like to thank the Director of Kenya Medical Research Institute (KEMRI) for the permission to publish this work.

**Contributors** MKN contributed to study design, did the formal analysis of the reported results, interpreted the results and wrote the first draft of the manuscript. PNM supervised the field team during data collection, was in charge of quality control and critically reviewed the manuscript. PM managed the study data and critically reviewed the manuscript. MK contributed to data collection and critically reviewed the manuscript. CRJCN contributed to study design, interpretation of the results and critically reviewed the manuscript. AAA conceptualised the study, acquired funding for the study, contributed to the study design, interpretation of the results and critically reviewed the manuscript. All the authors read and approved the final submitted version of the manuscript.

**Funding** This work was supported by funding from the Medical Research Council (Grant number MR/M025454/1) to AAA. This award is jointly funded by the UK Medical Research Council (MRC) and the UK Department for International Development (DFID) under MRC/DFID concordant agreement and is also part of the EDCTP2 program supported by the European Union. During this work, MKN was supported by the Wellcome Trust Master's Fellowship in Public Health and Tropical Medicine (Grant number 201310/Z/16/Z).

**Disclaimer** The funders did not have a role in the design and conduct of the original study.

**Competing interests** None declared.

**Patient consent** Not required.

**Ethics approval** This study was approved by the Kenya Medical Research Institute Scientific and Ethics Research Unit (Ref no: KEMRI/SERU/CGMR-C/030/3187). Permission to work in the comprehensive care clinic was also obtained from the Ministry of Health, Kilifi County government.

**Provenance and peer review** Not commissioned; externally peer reviewed.

**Data sharing statement** No additional data are available. Anyone interested in accessing the data reported in this article is free to write to the Data Governance Committee of the KEMRI Wellcome Trust Research Programme who will review the application and advise as appropriate, and ensure that uses are compatible with the consent obtained from participants for data collection. Requests can be sent to the coordinator of the Data Governance Committee using the following email: dgc@kemri-wellcome.org.

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
