## [Reviewer comments · BMJ Open]

ARTICLE DETAILS

TITLE (PROVISIONAL)	Neurocognitive and mental health outcomes and association with quality of life among adults living with HIV: A cross-sectional focus on a low-literacy population from Coastal Kenya
AUTHORS	Nyongesa, Moses; Mwangala, Patrick; Mwangi, Paul; Kombe, Martha; Newton, Charles; Abubakar, Amina A.

VERSION 1 – REVIEW

REVIEWER	Andrew Eaton Factor-Inwentash Faculty of Social Work, University of Toronto, Canada
REVIEW RETURNED	06-Jun-2018

GENERAL COMMENTS	This is a cross-sectional study comparing cognitive and mental health outcomes amongst people living with HIV and HIV-negative controls in Kenya. The article is well written and presents interesting findings that are appropriately discussed within the study's stated limitations. The background focuses nicely on the study's topic, with recent and relevant references throughout. The methods are clearly described and could be replicated, with especially strong detail regarding the measures. The statistics are well done and presented in an accessible manner. The discussion connects strongly with the background and results. Limitations are clearly stated and I appreciated the detail regarding whether community controls could potentially have been HIV-positive. Funding, ethical considerations, and STROBE checklist items are all transparently reported. Overall, a very interesting article that is presented well and makes a contribution to the field.
---

REVIEWER	Hetta Gouse University of Cape Town, South Africa
REVIEW RETURNED	19-Jun-2018

GENERAL COMMENTS	1. I do not see what reporting data on religion contributes to this study. Consider removing it.2. It is not clear what is meant by Digit Span Highest level reached. Please clarify.3. In addition to the Digit Span total score, please also report the mean number of digits that participants managed.4. Is there a reference for the parent study? If so, please include it.5. Regarding Limitations, please comment on the appropriateness of the neuropsychological tests used in this low level of education
--

VERSION 1 – AUTHOR RESPONSE

Reviewer #1

1. This is a cross-sectional study comparing cognitive and mental health outcomes amongst people living with HIV and HIV-negative controls in Kenya. The article is well written and presents interesting findings that are appropriately discussed within the study's stated limitations. The background focuses nicely on the study's topic, with recent and relevant references throughout. The methods are clearly described and could be replicated, with especially strong detail regarding the measures. The statistics are well done and presented in an accessible manner. The discussion connects strongly with the background and results. Limitations are clearly stated and I appreciated the detail regarding whether community controls could potentially have been HIV-positive. Funding, ethical considerations, and STROBE checklist items are all transparently reported. Overall, a very interesting article that is presented well and makes a contribution to the field.

We would like to thank the reviewer for the positive appraisal of this work.

Reviewer #2:

1. I do not see what reporting data on religion contributes to this study. Consider removing it.

We would like to thank the reviewer for this suggestion. We have now removed data reporting about religion.

2. It is not clear what is meant by Digit Span Highest level reached. Please clarify.

As a clarification, digit span highest level reached here referred to the highest set of digit length that participants managed to reach. The backward digit span was administered under 8 sets of digits of a given series length. Sets 1, 2, and 3 consisted of a series of 3 random digits between 1 and 10 (set 1 had 2 practice series of digits each with a digit length of 2). Sets 4, 5, 6, 7, and 8 also consisted of random digits between 1 to 10, each set having a series of digits of a length corresponding to the set, e.g. set 4 consisted of a series of digits with a digit length of 4. The computed mean was that of the highest set of digits reached by our participants.

To make this clearer, we now refer to this as “Highest set of digits reached” and elaborate what this means in the table legend (Table 2).

3. In addition to the Digit Span total score, please also report the mean number of digits that participants managed.

We computed two things from the backward digit span: 1) the mean total correct score; and 2) the mean of what we now refer to as the “highest set of digits reached”. The latter, in our opinion addresses this comment, since the test was administered in sets of digits with increasing level of length.

As explained in comment 2 above, the computed mean of the “highest set of digits reached” reflects the average level, in terms of increasing digit length set, that participants managed.

4. Is there a reference for the parent study? If so, please include it.

Currently, we do not have a published reference for the parent study as analysis and write-up is still ongoing.

5. Regarding Limitations, please comment on the appropriateness of the neuropsychological tests used in this low level of education population.

We would like to thank the reviewer for this comment. We agree that education level may have a partial role on neurocognitive test performance where more educated subjects are likely to perform better than the less educated due to differences in intellectual functioning. We recruited a low literacy sample (both adults living with HIV and community controls). As a result of this bias in selection, their cognitive performance on average, is expected to be lower when compared to populations with varying levels of educational achievement. Even though this bias applies equally for both study groups, the potential lack of familiarity with neurocognitive test requirement/demand restricts the generalization of our findings.

In the limitations section of our manuscript (page 15), we now include the following statement:

“Lastly, we recruited and administered neurocognitive tests to a low literacy sample. The potential lack of familiarity with neurocognitive test requirement/demands by our study participants, because of this biased selection, limits the generalizability of our finding. Our findings should be interpreted in the context of a low-literacy population.”